# Scabies in the Amhara region of northern Ethiopia: a cross-sectional study of prevalence, determinants, clinical presentation and community knowledge

Robel Yirgu ,[1,2,3] Jo Middleton ,[2,4,5] Abebaw Fekadu ,[2,3] Jackie A Cassell ,[2,5] Abraham Tesfaye,[3] Christopher Iain Jones ,[5] Stephen Bremner,[2,5] Wendemagegn Enbiale,[6] Gail Davey[1,2]

For numbered affiliations see end of article.

**Correspondence to**
Dr Robel Yirgu;
yirgurob@gmail.com

## ABSTRACT

**Background** The WHO aims to prevent, eliminate or control neglected tropical diseases, including scabies, by 2030. However, limited epidemiological data presented a challenge to control efforts, especially in high burden countries. There was a major scabies outbreak in northern Ethiopia starting in 2015 and prevalence has since increased across much of the country.

**Objective** To estimate scabies prevalence, identify its predictors, and assess community perception of, and knowledge about, the infestation.

**Design** Population-based cross-sectional study.

**Study setting** Ayu Guagusa district, Amhara region, northern Ethiopia.

**Participants** 1437 people who were members of 381 randomly selected households participated in the study. Five trained mid-level health workers clinically diagnosed people with scabies.

**Outcome measures** Clinically diagnosed scabies infestation.

**Data analysis** Multi-level logistic regression models were fitted to adjust for individual and household-level confounding variables, and identify predictors of scabies infestation.

**Results** Scabies prevalence was 13.4% (95% CI 11.8 to 15.5). Households of more than five people (adjusted OR (aOR)=3.5, 95% CI 1.2 to 10.2) were associated with increased odds of developing scabies, however, females had lower odds (aOR=0.5 95% CI 0.3 to 0.8). Scabietic lesions most frequently involved the trunk (62.0%), and vesicles were the most common types of lesions (67.7%). Two-thirds of adult study participants had heard about scabies and most obtained scabies related information from informal sources. Only 32% of cases sought care for scabies from any source.

**Conclusion** Scabies prevalence was high, signifying the need for community-based control interventions. Host density and sex were important predictors of scabies. Despite the favourable attitude toward the effectiveness of scabies treatment, healthcare seeking was low.

## STRENGTHS AND LIMITATIONS OF THE STUDY

⇒ The clinical diagnosis that we employed in this study cannot detect cases in the asymptomatic incubation period.

⇒ Physical examination of participants did not involve private body parts and we might have missed some people with scabies.

⇒ Single dose ivermectin-only mass drug administration against onchocerciasis had been underway in the study district. This could have treated scabies cases, undermining the true prevalence.

⇒ Though in our model, household size predicted increased odds of scabies, we could not objectively attribute this to host density, as we neither measured the size of housing units nor assessed sleeping arrangements.

by 2030.[1] Achieving this is contingent on availability of epidemiological data on NTD burden and distribution.[2] However, due to long standing neglect we know little about the epidemiology of many NTDs, including scabies, especially in high-burden low-income settings.[3]

Scabies is a skin infestation caused by the mite *Sarcoptes scabiei*,[4] which is established after a pregnant female burrows into the epidermis.[4 5] The average number of burrowing mites on an individual human host ranges between 10 and 15 but severe infestations of 'crusted scabies' (which are less common), can involve thousands of mites on an individual host.[6] Typical manifestations include generalised itch, rash and contact history with a presumed scabies patient.[7 8] Burrows are pathognomonic but rarely identified before thorough physical examination by experienced practitioners.[8] Presentation can also include pustules, nodules, chronic

## INTRODUCTION

The WHO aims to prevent, eliminate or control neglected tropical diseases (NTDs)

excoriation and skin crust.[9] Though itch and sleep deprivation are the primary complications of scabies, secondary bacterial infections are associated with cellulitis, fasciitis, septicaemia, acute glomerulonephritis and rheumatic heart disease.[6 10 11] Clinical diagnosis is the most frequently used approach in clinical and field settings, though accuracy relies heavily on examiners' knowledge and experience of scabies.[12]

Scabies accounts for an estimated 0.21% of disability-adjusted life-years, globally.[13] Prolonged physical contact is the main mode of scabies transmission, and circumstances that promote crowding predict increased transmission risk.[14 15] Outbreaks are thus common in institutional settings such as prisons, residential care homes for elderly people, hospitals and refugee camps.[15 16] Among individual level characteristics, age (children, adolescents and the elderly), low socioeconomic status and travel history to areas of higher scabies prevalence have been identified as increasing risk of developing scabies.[12 17] Facility-based studies rank scabies among the top three to five diseases in dermatology clinics.[18 19] On the other hand, community-based studies still indicate a pervasive need for scabies care and a significant delay to care seeking.[20 21] Limited geographic and financial access to healthcare services, lack of knowledge about the disease and treatment, and stigma and discrimination against scabies patients can contribute to untimely care seeking.[19 21 22]

Ethiopia reported a major scabies outbreak in 2015.[23] Studies conducted between 2015 and 2017 in the northern and southern parts of the country indicated a prevalence ranging between 11% and 33%.[23–26] This marked a significant increase from the 4% to 7% estimates in pre-outbreak facility-based studies.[26–28] Paucity of population-based data on the pattern and drivers of the outbreak has hindered the design of effective control interventions. In this study, we aimed to determine the epidemiology of scabies in Ayu Guagusa district and characterise the community's knowledge and perceptions of the infestation.

## METHODS
### Study design and setting
We conducted this cross-sectional study in Ayu Guagusa district, Amhara region, northern Ethiopia in December 2018. As well as determining scabies epidemiology and community knowledge and attitudes, it was designed to function as a baseline for longitudinal research measuring the secondary impact on scabies prevalence of single dose ivermectin-only mass drug administration (MDA).[29] The Ethiopian nation state is administratively divided into eleven regional states and two metropolitan councils, with regions subdivided into zones, districts and then *kebeles* (the lowest administrative unit comprising 3000–5000 people).[30] The state then organises households in each *kebele* into *Gotes*, small villages holding semi-official positions in the administrative structure. Amhara is the second most populous region with 17 million people,

85% of whom draw livelihoods from rain-dependent agriculture.[31 32] The Ethiopian state healthcare system is a three-tier system. Primary healthcare units provide disease prevention and health promotion and comprise a primary hospital, health centre and five satellite health posts. General hospitals provide secondary level care, with specialised and teaching hospitals serving as centres for tertiary specialised care. Health centre staff are mid-level health workers, trained to provide care for patients who do not require advanced medical attention. Health posts are the closest state health facilities to most communities, one post serves 3000–5000 people. Health posts are staffed with two health extension workers (HEWs), who provide primary healthcare services and facility patient referrals.[30]

### Sampling procedure
We selected northern Ethiopia for this study as we aimed to investigate the scabies outbreak in the region. Ayu Guagusa district (online supplemental figure S1) was purposively selected as it was the only district, among 11 other districts in Awi zone of Amhara region, in which MDA to eliminate onchocerciasis was underway. There are 21 *kebeles* in the district. Six were selected for this study by using simple random sampling. The number of kebeles is determined by dividing the sample size to the average size of kebele population (3000–5000). Then, one *gote* (small village) was randomly selected from each *kebele*. A sampling frame comprising a list of households in the selected *gotes* was prepared. Simple random sampling was employed to select study households from the frame. Eventually, all consenting members of selected households participated in the study.

### Data collection
Sociodemographic characteristics, healthcare seeking for scabies manifestations, contact history with a person showing scabies-suggestive symptoms, taking ivermectin tablets during the last round of MDA against onchocerciasis, and knowledge of and attitude toward scabies were measured at the level of the individual participants. Household wealth, household size and keeping domestic animals in the homestead were measured at the household level.

### Data collection instruments and procedures
An interview questionnaire was used to collect data from adult study participants, and parents or guardians of children aged <15 years (online supplemental table S1). After the tool was translated to the local language, *Amharic*, we pretested it with 12 people selected from a *kebele* near to the study district.

Clinical diagnosis was employed to identify scabies cases. Classic manifestations such as itch, skin rash with typical distribution and contact history with a person exhibiting scabies symptoms were considered.[12] Clinical examination with naked eye was carried out in sufficiently illuminated rooms in participants' residences. During

examination most parts of the body were exposed except the genitals and breasts. A chaperone was present for examinations of participants <18 years. Four nurses and one health officer, who were clinical staff in the nearby primary hospital, conducted the clinical examinations. These had prior experience identifying scabies patients in-clinics and were given a 3-day refresher training on scabies diagnosis by a dermatovenerologist (coauthor WE) from Bahir Dar University. The training aimed to standardise characterisation of disease manifestations and minimise interobserver bias in diagnosing scabies. Additionally, during the first few days of the field work, data collectors sent every participant identified as a scabies patient to the dermatologist. At times when there were disagreements in diagnosis, WE discussed the case with the data collectors and the consensual diagnosis was taken.

## Sample size

We considered an average household size of rural communities as four and intraclass correlation coefficient of 0.7[33] to calculate a design effect of 3.1. Including 15% non-participation rate we needed to involve 1210 people to detect 33% scabies prevalence with 5% precision. This sample size is also sufficient to fit multi-level logistic regression models with >10 variables while maintaining an event per variable ratio of >10.

## Data processing and analysis

An electronic template was prepared using Epidata V.3.01 (EpiData Association, Odense) statistical software to enter data from the paper-based questionnaire. Data were later exported to Stata V.14.0 (StataCorp LLC, Texas) for analysis. Summary measures were calculated for categorical data (frequency and percentage) and continuous data (mean and SD or median and IQR). Study households were categorised into wealth quintiles factoring in ownership of fixed household assets. Principal component analysis was used to generate components which best captured variability in wealth between the study households. Twenty-six binary variables of household asset and livestock ownership (including access to infrastructure) were included in the analysis (online supplemental table S2) and no significant correlation was observed among these covariates. Twenty-two principal components were generated with varimax rotation, and the first component, which had the highest *eigen value,* was used to generate the wealth index.

The dataset was hierarchical at two levels (ie, household and individual household member), and there was a high possibility for observations among members of the same household to be correlated. The intra-class correlation coefficient (ICC) was used to characterise the level of correlation between observations from the same household. The ICC was 0.88 (95% CI 0.79 to 0.94), indicating a significant cluster effect between members of the same study household, justifying the multi-level modelling approach. We thus fitted multi-level logistic regression

models to calculate the independent effects of individual and household level predictors on the odds of developing scabies. The models included individual participant level variables (ie, age, sex, marital status, level of education, occupation, contact history with a person exhibiting signs of scabies, bathing and participating in the last round of MDA) and three household level variables (ie, household size, presence of domestic animals in the homestead and household wealth) and a random effect for household.

We applied the *melogit* stata command to fit the models and calculate adjusted OR (aOR), with 95% CI to determine the strength and direction of association between the dependent and the predictor variables. Variable inflation factors (VIFs) were calculated to detect multicollinearity among the predictor variables, but none were dropped from the model as all the VIF values were less than 5.

## Patient and public involvement

People affected by podoconiosis, a non-infectious lymph-oedema caused by prolonged exposure of the feet to irritant soil, another heavily neglected tropical condition, were involved in the development of the funding proposal supporting this study. However, at that point in time (2016), there was no scabies patient association in existence in Ethiopia. Subsequent proposals have included input from people affected by scabies, through proposal development workshops and a newly established community of practice.

## RESULTS
### Participants

Across six *kebeles*, a total of 1437 people (797 females, 640 males) out of 1833 registered members of the 381 study households participated. Three hundred and ninety-six people were excluded after data collectors reported failure to find them after three subsequent visits to their households (online supplemental figure S2). All members of the selected households approached for recruitment consented to participate. This may surprise some readers. However, high consent rates are not uncommon in medical research in rural Ethiopian communities.[34 35] This may be explained by the community's respect for and trust of health workers and the frequent routine contact communities have with HEWs when they are doing outreach.[36]

Median age was 19 years (range, 10–37 years) (table 1). Almost half the participants (49.5%) were ≤18 years, 30.3% were 19–40 years, but, broadly in line with Ethiopia's demographics,[34] 20.3% were ≥41 years (compared with 16% of the national population). Most did not have a formal education (59.7%) and only 7% of participants had secondary level or above education. The most common occupational group was farming (52.2%), followed by being a student (47.8%). Most participants ≥16 years were married (38.8% of total participants). The number of households with a household size of <5

**Table 1** Socioeconomic characteristics of the study sample (n=1437)

| Variables | Categories | Overall (n=1437) n (%) | Scabies diagnosis | |
| --- | --- | --- | --- | --- |
| | | | No scabies manifestations (n=1245) n (%) | Clinical scabies (n=192) n (%) |
| Sex | Male | 640 (44.5) | 539 (43.3) | 101 (52.6) |
| | Female | 797 (55.5) | 706 (56.7) | 91 (47.4) |
| Age in years | ≤10 | 365 (25.4) | 316 (25.4) | 49 (25.5) |
| | 11–18 | 346 (24.1) | 287 (23.1) | 59 (30.7) |
| | 19–40 | 435 (30.3) | 387 (31.1) | 48 (25.0) |
| | ≥41* | 291 (20.3) | 255 (20.5) | 36 (18.8) |
| | Median age (IQR) | 19 (10–37) | 20 (10–38) | 16 (10–30) |
| Level of education† (n=1422) | No formal education | 849 (59.7) | 745 (60.6) | 104 (54.2) |
| | Primary education | 474 (33.3) | 401 (32.6) | 73 (38.0) |
| | Secondary and above | 99 (7.0) | 84 (6.8) | 15 (7.8) |
| Occupation (n=1412) | Farmer | 737 (52.2) | 650 (53.3) | 87 (45.3) |
| | Student | 675 (47.8) | 570 (46.7) | 105 (54.7) |
| Marital status‡ | Not married | 228 (15.9) | 204 (16.4) | 24 (12.5) |
| | Married | 558 (38.8) | 487 (39.1) | 71 (37.0) |
| | <16 years | 651 (45.3) | 554 (44.5) | 97 (50.5) |
| Household size | <5 people | 707 (49.2) | 637 (51.2) | 70 (36.5) |
| | ≥5 people | 730 (50.8) | 608 (48.8) | 122 (63.5) |
| Contact history with symptomatic person | No | 947 (65.9) | 939 (75.4) | 8 (4.2) |
| | Yes | 490 (34.1) | 306 (24.6) | 184 (95.8) |
| Took pill during the last mass drug administration | No | 338 (23.5) | 274 (22.0) | 64 (33.3) |
| | Yes | 1099 (76.5) | 971 (78.0) | 128 (66.7) |
| Domestic animals kept in the homestead | No | 202 (14.1) | 171 (13.7) | 31 (16.2) |
| | Yes | 1235 (85.9) | 1074 (86.3) | 161 (83.9) |
| Household wealth | Lowest | 291 (20.3) | 266 (21.4) | 25 (13.0) |
| | Second | 277 (19.3) | 240 (19.3) | 37 (19.3) |
| | Middle | 293 (20.4) | 255 (20.5) | 38 (19.8) |
| | Fourth | 293 (20.4) | 261 (20.9) | 32 (16.7) |
| | Highest | 283 (19.7) | 223 (17.9) | 60 (31.3) |
| Main source of water for any purpose§ | Public tap | 181 (12.6) | 139 (11.3) | 42 (21.9) |
| | Dug well | 405 (28.2) | 355 (28.5) | 50 (26.0) |
| | Spring water | 850 (59.2) | 746 (59.9) | 104 (54.2) |
| Frequency of bathing per week | <1 | 382 (26.6) | 326 (26.2) | 56 (29.2) |
| | 1–2 | 789 (54.9) | 688 (55.3) | 101 (52.6) |
| | ≥3 | 266 (18.5) | 231 (18.6) | 35 (18.2) |
| Frequency of washing clothes per week | <1 | 199 (13.9) | 174 (13.9) | 25 (13.0) |
| | 1–2 | 1116 (77.7) | 967 (77.7) | 149 (77.6) |
| | ≥3 | 122 (8.5) | 104 (8.4) | 18 (9.4) |

*People >40 years comprises just 16% of the Ethiopian population.
†This category comprised people with no formal education and children younger than school age.
‡Marital status of participants was only asked of those >16 years.
§Total scores exceed 100% as individual participants could select multiple response categories.

(49.2%) and ≥5 (50.8%) was similar. Most study households (70.6%) always used surface water for any purpose. 26.6% of participants reported bathing less than weekly, 54.9% reported bathing 1–2 times weekly, and 18.5% ≥3 times weekly. 13.9% of interviewees stated their clothes were washed less than once per week, 77.7% 1–2 times weekly and 8.5% ≥3 times weekly. Median time for a round trip to water source was 20 min.

## Scabies cases

Scabies prevalence was 13.4% (95% CI 11.8% to 15.5%). The highest age specific prevalence was among people aged 11–18 years 17.1% (95% CI 13.1% to 21.0%) and the lowest among those between 31 and 40 years 6.7% (95% CI 2.9% to 10.6%). The prevalence among people in the extremes of age: 0–10 years and people older than 41 years was 13.4% (95% CI 9.9% to 16.9%) and 12.4% (95% CI 8.6% to 16.2%), respectively. The socioeconomic characteristics and scabies diagnoses of those examined are provided in table 1. One hundred and ninety-two participants had scabies (101 females, 91 males) (online supplemental table S3). Participants with clinical scabies were younger (median 16 years, IQR 10–30 years) than those without signs and symptoms (median 20 years, IQR 10–38 years). Those with no scabies manifestations were evenly split between households with <5 and ≥5 people (respectively: 707, 49.2%; 730, 50.8%), whereas a markedly higher proportion of scabies cases lived in households with ≥5 people (122, 63.5%) than <5 (70, 36.5%). Stated frequency of clothes washing and personal bathing were similar between the scabies cases and those without manifestations.

Clinical presentations of identified scabies cases (n=192) are summarised in online supplemental table S4. Vesicles were the most frequently found clinical signs (67.7%). The trunk and head were the most (62.0%) and least (3.6%) involved body parts. Nearly all individuals diagnosed with scabies (184, 95.8%) reported experiencing itch and the majority (165, 85.9%) stated they had a contact history with a presumed scabies case, most stating they believed they had become infested due to contact with a household member (146, 80.7%). Out of the 192 clinical scabies cases, only 62 (32.3%) reported seeking care by the time the data were collected. Of those that did, the majority (40, 64.5%) reported seeking care from a health facility, while a smaller proportion had carried out self-treatment (14, 22.6%). Three (4.8%) said they had used holy water as a remedy. The median number of days between the onset of symptoms and seeking treatment was 20 (IQR 7–30) (online supplemental table S5). Twenty-seven participants said they sought treatment later than this, giving the following reasons: 21 (77.8%) because they assumed the signs and symptoms would disappear on their own, 3 (11.1%) due to lack of time to seek care, 2 (7.1%) believed medical expenses would be high and 2 (7.1%) because they considered the symptoms mild.

## Community knowledge and attitudes about scabies

The knowledge and attitudes of adult study participants are summarised in tables 2 and 3. Two-thirds (n=500) reported they had heard about scabies. When asked how they believed scabies was transmitted, most stated beliefs in line with established scientific knowledge: prolonged physical contact (57.6%); sharing clothes and bedding with a scabies patient (51.4%); sharing sleeping space with a scabies patient (42.4%). However, some stated modes of transmission not in line with scientific understanding: via contaminated water (10.0%); via blood contact (4.0%); from the soil (2.8%). Most said they wanted to know more about the disease, with a higher proportion interested among the scabies cases. When asked to choose from a list, most (93%) selected body creams and anti-scabies tablets as scabies treatments, while small minorities chose holy water (4.4%), traditional treatment (home remedies and traditional herbs) (3.4%) or prayer (1.4%). 56.3% of adults without scabies said they thought any community member could acquire scabies (table 2). Nearly all (441, 95.7%) said they would seek care from healthcare facilities if they developed scabies (table 3). Two hundred and thirteen (48.3%) reported they would feel nothing emotionally if they developed scabies, but 116 (26.3%) said they would feel shame, and 113 (25.6%) fear. When asked to choose from a list of potential attitudes towards scabies patients, a similar proportion chose a response stating, 'they felt compassion and wanted to help' and 'they feared they would be infected by them' (respectively, 115, 56.4% and 104, 51.0%). One hundred and sixty-six (81.4%) chose a response stating, 'they believed their community would prefer to avoid scabies patients', and 46 (22.5%) that 'people would appear friendly but wish to avoid them' (table 3).

## Odds of infestation

Table 4 gives the results of the multi-level logistic regression analysis of determinants of scabies infestation. The odds of scabies infestation was lower among females (aOR=0.5, 95% CI 0.3 to 0.8), and people from households of five or more had higher odds of infestation (aOR=3.5, 95% CI 1.2 to 10.2). The following were not predictive of scabies diagnosis: occupation; education; marital status; age; frequency of bathing; taking ivermectin tablets during prior MDA; keeping domestic animals in the homestead; household wealth.

## DISCUSSION

Scabies prevalence in Ayu Guagusa district was high. Vesicles, excoriations and papules, often involving the trunk, were the most frequently observed signs, and nearly all participants complained of generalised itch. Only around a third of clinical scabies cases had sought care, with the median number of days before seeking care for the minority who did being 20 days. More than two-thirds had heard about scabies, and friends and family were the most common sources of scabies-related information. Most believed scabies was transmitted through prolonged physical contact with a scabies patient and that it could be treated with drugs. In the multi-level logistic regression model being from households of size >5 people increased odds of scabies infestation but female sex was associated with lower odds of scabies.

There are few studies on scabies in Africa, including Ethiopia, and population-based studies employing primary data are even rarer.[20 23 25 37 38] Scabies prevalence

**Table 2** Knowledge of adult study participants about scabies (n=500)

| Variable | Categories | Overall n (%)* | Scabies diagnosis among adults | |
| --- | --- | --- | --- | --- |
| | | | No scabies manifestations (n=430) n (%) | Clinical scabies (n=70) n (%) |
| From what sources first heard about scabies | Family and friends | 320 (64.0) | 269 (62.6) | 51 (72.9) |
| | Healthcare providers | 71 (14.2) | 65 (15.1) | 6 (8.6) |
| | Health extension workers | 48 (9.6) | 9 (2.9) | 39 (55.7) |
| | Mass media | 44 (8.8) | 33 (7.7) | 11 (15.7) |
| | Religious leaders | 18 (3.6) | 18 (4.2) | 0 |
| | Not collected | 44 (8.8) | 42 (9.8) | 2 (2.9) |
| What are the manifestations of scabies? | Itch | 449 (89.8) | 382 (88.8) | 67 (95.7) |
| | Rash | 242 (48.4) | 187 (43.5) | 55 (78.6) |
| | Skin crust | 24 (4.8) | 17 (3.9) | 7 (10.0) |
| How is scabies transmitted? | Prolonged physical contact with a scabies patient | 288 (57.6) | 245 (56.9) | 43 (61.4) |
| | Sharing clothing and bedding with a scabies patient | 257 (51.4) | 216 (50.2) | 41 (58.6) |
| | Sharing sleeping space with scabies patient | 212 (42.4) | 175 (40.7) | 37 (52.9) |
| | From contaminated water | 50 (10.0) | 49 (11.4) | 1 (1.4) |
| | Blood contact with a scabies patient | 20 (4.0) | 18 (4.2) | 2 (2.9) |
| | From the soil | 14 (2.8) | 12 (2.8) | 2 (2.9) |
| | No response | 58 (11.6) | 49 (11.4) | 9 (12.9) |
| Is scabies treatable? | Yes | 450 (90.0) | 387 (90.0) | 63 (90.0) |
| | No | 45 (9.0) | 39 (9.1) | 6 (8.6) |
| | No response | 5 (1.0) | 4 (1.0) | 1 (1.4) |
| What is the treatment for scabies? | Body creams and tablets | 469 (93.8) | 404 (93.9) | 65 (92.8) |
| | Holy water | 22 (4.4) | 21 (4.9) | 1 (1.4) |
| | Traditional treatment | 17 (3.4) | 16 (3.7) | 1 (1.4) |
| Who can catch scabies? | Any one | 282 (56.3) | 235 (54.5) | 47 (67.1) |
| | Poor people | 53 (10.6) | 42 (9.7) | 11 (15.7) |
| | Rich people | 14 (2.8) | 11 (2.6) | 3 (4.3) |
| | Homeless people | 24 (4.8) | 20 (4.6) | 4 (5.7) |
| | Urban people | 14 (2.8) | 12 (2.8) | 2 (2.9) |
| | Rural people | 49 (11.1) | 38 (10.1) | 11 (16.9) |
| | Commercial sex workers | 22 (4.4) | 19 (4.4) | 3 (4.3) |
| | Prisoners | 22 (4.4) | 18 (4.2) | 4 (5.7) |
| | No response | 44 (8.8) | 39 (9.1) | 5 (7.1) |
| Do you wish to know more about scabies? | Yes | 435 (87.0) | 371 (86.3) | 64 (91.4) |
| | No | 65 (13.0) | 59 (13.7) | 6 (8.6) |

*Total scores may exceed 100% for some questions when individual participants could select more than one response categories.

estimates for different parts of Ethiopia range from 2.5% to 33.5%.[23] However, since most of these studies were conducted either in areas of scabies outbreaks or were secondary analyses on data related with scabies outbreak response activities, their prevalence estimates may not be representative of the remaining parts of the country.[23 39 40] In our study, estimated scabies prevalence was 13.4%. We believe this high community prevalence is likely to be, in part, related to the wider ongoing scabies outbreak across Amhara region, first reported in 2015.[26] The prevalence is comparable to studies in southern Ethiopia[25 37] but lower than estimates from studies elsewhere in the northern

parts of the country.[23 26 41] This may be a secondary impact of prior single dose ivermectin-only onchocerciasis MDA that had taken place 8 months before the current study.

Only around a third of those diagnosed with scabies had reported seeking care of any kind by the time this data were collected. Similar poor health seeking behaviour have been reported in other studies.[18 22] Not seeking care early because of low perceived severity of manifestations, was also found in a study in a Brazilian urban slum.[18] Among those who sought care, more than two-thirds reported attending a primary healthcare facility. Even though we could not rule out the effect

**Table 3** Attitude of adult study participants without scabies symptoms towards scabies (n=441)

| Question | Categories | n (%)* |
|---|---|---|
| Do you think you can acquire scabies? | Yes | 265 (61.0) |
| | No | 169 (38.3) |
| | No response | 7 (1.6) |
| What would you feel if you developed scabies? | Nothing | 213 (48.3) |
| | Shame | 116 (26.3) |
| | Fear | 113 (25.6) |
| | Disappointment | 68 (15.4) |
| | Surprise | 16 (3.6) |
| | No response | 13 (2.9) |
| What would you do if you developed scabies? | Seek care from healthcare facilities | 422 (95.7) |
| | Use traditional medicine | 2 (0.5) |
| | Do nothing because it improves on its own | 2 (0.5) |
| | No response | 16 (3.6) |
| What is your feeling towards scabies patients? (n=204)† | I feel compassion and wish to help | 115 (56.4) |
| | I fear they may infect me | 104 (51.0) |
| | I do not feel anything | 66 (32.4) |
| | I feel compassion but prefer to stay away | 40 (19.4) |
| | No response | 6 (2.9) |
| How does the community treat scabies patients? (n=204)† | Most people prefer to avoid scabies patients | 166 (81.4) |
| | People help and support them | 78 (38.2) |
| | People appear friendly but they wish they avoided them | 46 (22.5) |
| | No response | 25 (22.3) |

*All items in this table were multiselect and participants could give more than one response for each variable, and this might cause the cumulative score of some variables to exceed the total number of participants.
†Calculated out of 204 participants who have knowledge of a scabies patient.

**Table 4** Multi-level logistic regression analysis results of determinants of scabies infestation

| Variable | Category | Adjusted OR | 95% CI |
|---|---|---|---|
| Sex | Male (ref) | | |
| | Female | 0.5 | 0.3 to 0.8 |
| Age (years) | ≤17 (ref) | | |
| | 18–40 | 0.5 | 0.0 to 4.4 |
| | ≥41 | 0.4 | 0.0 to 4.2 |
| Occupation | Farmer (ref) | | |
| | Student | 0.9 | 0.3 to 3.2 |
| Education | No formal education (ref) | | |
| | Primary education | 1.1 | 0.5 to 2.3 |
| | Secondary and higher | 2.8 | 0.8 to 9.9 |
| Marital status | Not married (ref) | | |
| | Married | 0.8 | 0.1 to 7.0 |
| | <16 | 1.3 | 0.5 to 3.8 |
| Frequency of bathing per week | <1 (ref) | | |
| | 1–2 times | 0.6 | 0.3 to 1.4 |
| | ≥3 times | 1.0 | 0.3 to 2.9 |
| Took ivermectin during last mass drug administration | No (ref) | | |
| | Yes | 0.7 | 0.2 to 3.0 |
| Household size | <5 people (ref) | | |
| | ≥5 people | 3.5 | 1.2 to 10.2 |
| Domestic animals kept in homestead | No (ref) | | |
| | Yes | 0.7 | 0.2 to 3.0 |
| Household wealth | Lowest (ref) | | |
| | Second | 0.6 | 0.1 to 2.7 |
| | Middle | 0.8 | 0.2 to 4.1 |
| | Fourth | 0.8 | 0.2 to 4.2 |
| | Highest | 3.5 | 0.8 to 15.1 |
| Intracluster correlation coefficient (ICC, 95% CI) | | 0.89 | 0.79 to 0.94 |

Number of households, N=381; number of participants, N=1403; number of cases 192.

of social desirability bias (since the data collectors were health workers), the improving physical and financial access to primary health services could have made healthcare facilities the preferred source of care.[42] However, services at these primary healthcare facilities may benefit from enhancing quality of care and employing teledermatology.[43] Participants reported to the first author that frequent stock-outs of the commonly used cream (permethrin) in many health centres discouraged their attendance. Self-treatment was the second most frequently mentioned source of care. From our anecdotal field observation, people applied local herbs on the lesions.

Nearly two-thirds of the study population had heard about scabies, primarily from family and friends. The now pervasive awareness about scabies is, we suspect, mainly a result of its occurrence as an outbreak. More than half of the study participants in both groups (ie, clinical scabies and no manifestations of scabies) correctly identified 'prolonged physical contact with a scabies patient' as a mode of scabies transmission. There remains room

for educational interventions, as over a third in each group did not identify this mode of transmission. Local health worker teaching on scabies has focused heavily on personal hygiene. However, reported frequency of bathing was not predictive in our study, and experimental and epidemiological evidence from elsewhere indicates personal body washing is probably unrelated to scabies prevalence.[44] A better focus would be to prevent prolonged physical contact with presumed scabies cases (eg, temporary separation of sleeping arrangements for symptomatic individuals), and to treat all contacts once a scabies case is identified. Most participants believed there is treatment for scabies and a comparable number wanted

to know more about scabies in general. This presents an opportunity for effective behavioural change communications. Even though we did not use a validated tool to measure attitude, a subtle stigma was associated with scabies patients. This finding was consistent with Trettin *et al*, where scabies patients reported shame and fear of stigma as scabies was perceived to be associated with poor hygiene and poverty.[45 46] Arguably, the present focus on personal hygiene in teachings surrounding scabies may risk contributing to such social stigma against scabies patients.

High scabies prevalence is reported among children in high scabies burden settings, and in a global systematic review by Romani *et al*[47] prevalence was lower in adults than in children in all relevant studies. In our study, though the association between age and scabies was not statistically significant, the majority of the scabies cases were still among people aged <18 years. Household size was a strong predictor; odds of scabies infestation among people living in households with ≥5 members (a household size larger than the national average)[48] was high. Similarly, Hegazy *et al*[49] reported that people from households of high crowding index had higher odds of scabies infestation in rural Egypt. Since prolonged physical contact is required for effective scabies transmission, high host density both in households and institutional settings facilitates transmission.[50]

## Conclusion

Scabies prevalence in Ayu Guagusa district is high and crosses the recommended threshold to initiate preventive chemotherapy.[51] MDA needs to be initiated as the outbreak can no longer be controlled through standard care.[52] Ivermectin MDAs for scabies were first trialled in Oceania in the late 1990s and have proven to be effective at reducing community prevalence.[53] Ideally two doses of oral ivermectin (200 µg/kg) should be given 7–14 days apart, with alternative scabicides provided to whom ivermectin is presently not recommended (children weighing <15 kg or <90 cm height; pregnant women; women breast feeding in the first week). MDA rounds should have a minimum of 80% population coverage and conducted annually for 3–5 years, with a stopping threshold of 2% community prevalence.[51] Accompanying such MDAs with educational interventions may also help to: (1) increase awareness about modes of scabies transmission; (2) address the subtle stigma attached to scabies and (3) support communities to modify environmental circumstances that facilitate transmission.

## Author affiliations
[1]School of Public Health, Addis Ababa University College of Health Sciences, Addis Ababa, Ethiopia
[2]NIHR Global Health Research Unit on Neglected Tropical Diseases, Brighton and Sussex Medical School, Falmer, UK
[3]Centre for Innovative Drug Development and Therapeutic Trials for Africa (CDT-Africa), Addis Ababa University, Addis Ababa, Ethiopia
[4]Department of Global Health and Infection, Brighton and Sussex Medical School, Brighton, UK
[5]Department of Primary Care and Public Health, Brighton and Sussex Medical School, Falmer, UK
[6]College of Medical and Health Sciences, Bahir Dar University, Bahir Dar, Ethiopia

**Acknowledgements** We thank the field team, healthcare workers from Gimjabet hospital, for conducting data collection and Dr Vasso Anagnostopoulou for supporting data management activities. Awi Zone and Ayu Guagusa District Health office administration provided invaluable background information about the institutions in the health system and for facilitating the field work. We are also grateful to the participants who welcomed our field team into their homes and gave their time for the interview.

**Contributors** This study was carried out as part of the work of the NIHR Global Health Research Unit on Neglected Tropical Diseases Phase 1 at Brighton and Sussex Medical School. For clarity we detail contributions using the CRediT Contributor Roles Taxonomy (https://casrai.org/credit/) and provide employment and disciplinary descriptions. Conceptualisation: JM, AF, JAC, GD. Analysis: RY, JM. Investigation: RY, AT, WE. Methodology: RY, JM, AF, JAC, SB, CIJ, GD. Supervision: JM, AF, JAC, GD. Writing – original draft: RY, JM. Visualisation: JM, AT. Writing – review and editing: RY, JM, AF, JAC, AT, SB, CIJ, WE, GD. RY accepts full responsibility for the work and/or the conduct of the study, had access to the data, and controlled the decision to publish.

**Funding** 131996). The views expressed are those of the authors and not necessarily those of the NHS, the NIHR or the Department of Health and Social Care.131996). The views expressed are those of the authors and not necessarily those of the NHS, the NIHR or the Department of Health and Social Care.131996). The views expressed are those of the authors and not necessarily those of the NHS, the NIHR or the Department of Health and Social Care.

**Competing interests** None declared.

**Patient and public involvement** Patients and/or the public were not involved in the design, or conduct, or reporting, or dissemination plans of this research.

**Patient consent for publication** Consent obtained directly from patient(s).

**Ethics approval** The study involved human participants and ethics approval was obtained from Addis Ababa University Institutional Review Board (IRB) (AAUMF 03-008) and Brighton and Sussex Medical School (BSMS) Research Governance and Ethics Committee (RGEC) (ER/BSMS9G1Z/1). Permission was sought from the Ministry of Health (MoH), Amhara regional health Bureau and local administrative heads before commencing field work. Informed consent was requested and received from all adult participants, and assent from minors 15–17 years. Minors <15 years participated in the study once their parents or guardians gave consent. Participants diagnosed with scabies or other diseases were referred to the nearest health centre for further investigation and treatment.

**Provenance and peer review** Not commissioned; externally peer reviewed.

**Data availability statement** Data are available upon reasonable request. All data relevant to the study are included in the article or uploaded as supplementary information.

**ORCID iDs**
Robel Yirgu http://orcid.org/0000-0002-8013-6172
Jo Middleton http://orcid.org/0000-0001-5951-6608
Abebaw Fekadu http://orcid.org/0000-0003-2219-0952

Jackie A Cassell http://orcid.org/0000-0003-0777-0385
Christopher Iain Jones http://orcid.org/0000-0001-7065-1157

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
