## [Reviewer comments · BMJ Open]

This paper was submitted to a another journal from BMJ but declined for publication following peer review. The authors addressed the reviewers' comments and submitted the revised paper to BMJ Open. The paper was subsequently accepted for publication at BMJ Open.

ARTICLE DETAILS

TITLE (PROVISIONAL)	Scabies in the Amhara region of northern Ethiopia: a cross-sectional study of prevalence, determinants, clinical presentation, and community knowledge
AUTHORS	Yirgu, Robel; Middleton, Jo; Fekadu, Abebaw; Cassell, Jackie; Tesfaye, Abraham; Jones, Christopher; Bremner, Stephen; Enbiale, Wendemagegn; Davey, Gail

VERSION 1 – REVIEW

REVIEWER	Joshi, Tejas P. Baylor College of Medicine
REVIEW RETURNED	14-Jul-2023

GENERAL COMMENTS	Thank you for this excellent article assessing the prevalence of scabies in Ethiopia. There are very little data on this topic, and this article represents a valuable contribution to scholarship. Please see below for my comments: 1. Lines 141-142: "During examination most parts of the body were exposed except the genitals." The authors may wish to acknowledge this as a limitation of the study as scabies can also often involve the genitalia. Potential cases of scabies limited to the genital region may have been missed.2. Were mineral oil scrapings done to confirm scabies infection? If not, this should be acknowledged as a limitation of the manuscript.3. Line 348: authors may wish to cite the manuscript by Hotez and colleagues that discusses the utility of the rapid impact package in tackling sNTDs (please see PMID 24671756)4. Authors may also wish to comment on the utility of teledermatology in managing sNTDs like scabies (please see PMID 34631272)
---

REVIEWER	Gramp, Prudence Gold Coast University Hospital
REVIEW RETURNED	25-Jul-2023

GENERAL COMMENTS	The authors have identified a gap in the literature and their study addresses attempts to increase knowledge in this area. Overall a well written paper with some good considerations for confounders and study limitations.
--

	For a cross sectional study, I believe that the measure of effect should be presented as a prevalence ratio or a risk difference rather than an odds ratio. It is worth acknowledging there may be an underestimated number of positive cases due to not examining the genitals/groin of patients as this is a common site for scabies, scabies nodules pathognomonic etc. Breasts can also be a common site for scabies. Limitations of clinical examinations could be expanded in the discussion. Page 7, line 187 – It would be worth stating what podoconiosis is, eg. podoconiosis (a non-filarial elephantitis: an inflammatory reaction to mineral soil particles) For table 1 – provide a simple key for how data was presented eg n (%)
--	--

REVIEWER	Ampem Amoako, Yaw Kwame Nkrumah University of Science and Technology
REVIEW RETURNED	30-Jul-2023

GENERAL COMMENTS	The study objectives and design are clear. the reults have been well presented and the findings clearly explained. I commend the authors for undertaking this important study on this very relevant disease among the population of Ethiopia.
---

VERSION 1 – AUTHOR RESPONSE

Reviewer#1

1. Lines 141-142: "During examination most parts of the body were exposed except the genitals." The authors may wish to acknowledge this as a limitation of the study as scabies can also often involve the genitalia. Potential cases of scabies limited to the genital region may have been missed.

Authors' response: We concur with this comment. We explained the implications of excluding the genitals from the clinical examination in the strengths and limitations section (page-3).

2. Were mineral oil scrapings done to confirm scabies infection? If not, this should be acknowledged as a limitation of the manuscript.

Authors' response: We agree with the reviewer that using microscopy would have increased the specificity of our diagnosis. However, the low sensitivity of microscopy could lead to misclassifying true scabies cases (Cassell JA et.al., 2018). Apart from the potential misclassification bias it might as well undermine the power of our analysis by decreasing the number of cases among the study sample. For this reason, we have decided to employ the second level of diagnostic certainty (clinical scabies), where the first level is microscopy confirmed scabies, as indicated in the IACS Delphi consensus criteria (please refer PMID: PMC7687112).

3. Line 348: authors may wish to cite the manuscript by Hotez and colleagues that discusses the utility of the rapid impact package in tackling sNTDs (please see PMID 24671756).

Authors' response: Thank you for bringing this important article to our attention, we have cited it in our manuscript (page-13).

4. Authors may also wish to comment on the utility of teledermatology in managing sNTDs like scabies (please see PMID 34631272)

Authors' response: We agree that teledermatology is an important tool to ensure universal access to care for sNTDs. We have cited this important article to support our argument on the need to strengthen primary health care facilities to provide comprehensive care for sNTDs.

Reviewer#2

5. For a cross sectional study, I believe that the measure of effect should be presented as a prevalence ratio or a risk difference rather than an odds ratio.

Authors' response:

We thank the reviewer for this remark. All three measures of effect are valid in a cross-sectional study but the natural one from a multilevel logistic regression model is the odds ratio, which we adjusted for the variables listed in the methods section. To estimate the adjusted prevalence ratio, which does not have the same convenient property of reciprocity as the odds ratio (Tamhane et al. 2016), we would need to have instead attempted to fit a log-binomial model using generalized estimating equations or a generalized linear model with log link, binomial family and standard errors adjusted for clustering by household. Given the fairly large number of independent variables in the model, non-convergence would likely be a problem with each of these. Risk differences can be estimated similarly (binomial family, identity link) but the models are subject to the same potential for non-convergence problems. Finally, robust Poisson regression is an alternative approach to obtain estimates of prevalence ratios or risk differences but can result in confidence intervals that are too wide (Holmberg and Anderson 2020). Therefore, we prefer to report odds ratios.

6. It is worth acknowledging there may be an underestimated number of positive cases due to not examining the genitals/groin of patients as this is a common site for scabies, scabies nodules pathognomonic etc. Breasts can also be a common site for scabies. Limitations of clinical examinations could be expanded in the discussion.

Authors' response: This is a valid point and we have addressed this limitation, please refer response number-1 in this document.

7. Page 7, line 187 – It would be worth stating what podoconiosis is, eg. podoconiosis (a non-filarial elephantitis: an inflammatory reaction to mineral soil particles).

Authors' response: Thank you for pointing this out. We have included a sentence defining podoconiosis (page-7).

8. For table 1 – provide a simple key for how data was presented eg n (%).

Authors' response: A column descriptor is included in all the three columns of Table-1.

VERSION 2 – REVIEW

REVIEWER	Joshi, Tejas P. Baylor College of Medicine
REVIEW RETURNED	21-Aug-2023

GENERAL COMMENTS	Thank you very much for your revisions.
---

REVIEWER	Gramp, Prudence Gold Coast University Hospital
REVIEW RETURNED	10-Sep-2023

GENERAL COMMENTS	An important study to add to the literature.
--